# Invasive Candidiasis in Contexts of Armed Conflict, High Violence, and Forced Displacement in Latin America and the Caribbean (2005–2025)

**DOI:** 10.3390/jof11080583

**Published:** 2025-08-06

**Authors:** Pilar Rivas-Pinedo, Juan Camilo Motta, Jose Millan Onate Gutierrez

**Affiliations:** 1Medical and Diagnostic Mycology Group, Department of Microbiology, Faculty of Medicine, Universidad Nacional de Colombia, Bogotá 111321, Colombia; 2Infectious Disease Service, Universidad Nacional de Colombia, Bogotá 111321, Colombia; 3Internal Medicine Service, Fundación Cardioinfantil–Instituto de Cardiología, Bogotá 110131, Colombia; 4Infectious Diseases Service, Clínica Sebastian de Belalcázar, Clínica Colsanitas, Keralty Group, Cali 760042, Colombia; 5Infectious Diseases Service, Clínica Imbanaco, Cali 760042, Colombia; 6Infectious Diseases Service, Clínica de Occidente S.A., Cali 760042, Colombia

**Keywords:** invasive candidiasis, candidemia, armed conflict, forced displacement, Latin America and the Caribbean, antifungal resistance, vulnerable populations, epidemiological surveillance

## Abstract

Invasive candidiasis (IC), characterized by the most common clinical manifestation of candidemia, is a fungal infection with a high mortality rate and a significant impact on global public health. It is estimated that each year there are between 227,000 and 250,000 hospitalizations related to IC, with more than 100,000 associated deaths. In Latin America and the Caribbean (LA&C), the absence of a standardized surveillance system has led to multicenter studies documenting incidences ranging from 0.74 to 6.0 cases per 1000 hospital admissions, equivalent to 50,000–60,000 hospitalizations annually, with mortality rates of up to 60% in certain high-risk groups. Armed conflicts and structural violence in LA&C cause forced displacement, the collapse of health systems, and poor living conditions—such as overcrowding, malnutrition, and lack of sanitation—which increase vulnerability to opportunistic infections, such as IC. Insufficient specialized laboratories, diagnostic technology, and trained personnel impede pathogen identification and delay timely initiation of antifungal therapy. Furthermore, the empirical use of broad-spectrum antibiotics and the limited availability of echinocandins and lipid formulations of amphotericin B have promoted the emergence of resistant non-*albicans* strains, such as *Candida tropicalis*, *Candida parapsilosis*, and, in recent outbreaks, *Candidozyma auris*.

## 1. Introduction

Invasive candidiasis (IC), which includes both candidemia and other deep forms of *Candida* infection, is among the most common hospital-acquired fungal infections. A high disease burden and mortality characterize it, often linked to prolonged hospitalizations and the coexistence of various comorbidities, thus posing a major challenge to healthcare systems globally [1]. Over the past twenty years, there has been a steady increase in the number of individuals at risk of developing IC, driven by the growing use of immunosuppressive therapies, stays in intensive care units (ICUs), and frequent administration of broad-spectrum antibiotics. This situation is further complicated by the emergence of non-*albicans* strains that are resistant to antifungal agents, making it difficult to treat these infections effectively [2]. Empirical and prolonged antibiotic use in critically ill patients fosters antifungal-resistant *Candida* (e.g., *C. tropicalis*, *C. parapsilosis*) and, coupled with limited echinocandins, drives emergence of non-*albicans* strains. In recent years, outbreaks of the emerging isolates *C. auris* have also been documented, characterized by its resistance and difficult control in hospital settings [1,3,4].

It is estimated that, worldwide, IC causes around 227,300 hospitalizations each year, with at least 100,000 confirmed deaths. However, if unreported cases and those not diagnosed in time are included, estimates could rise to 1,565,000 episodes and nearly 995,000 deaths, underscoring its relevance as a critical public health priority [5,6,7]. In the LA&C region, the timely detection of IC faces multiple obstacles, including a high concentration of risk factors and restricted access to effective antifungal treatments. This situation is estimated to cause around 50,000 hospitalizations annually and between 15,000 and 35,000 deaths attributable to the infection [1,8].

Over the past two decades, many countries in LA&C have experienced prolonged crises linked to armed conflict, structural violence, and forced displacement. These situations have significantly deteriorated health systems, leading to drug shortages and severe limitations in medical care. It has been documented that these conditions significantly increase the susceptibility of affected populations to invasive fungal infections (IFI), including IC [5,9,10]. It is estimated that more than 1.6 million deaths worldwide each year are the result of serious fungal diseases, with areas affected by forced displacement having the highest mortality rates [5,10,11]. Despite the significant regional burden of candidiasis, there are still no specific studies measuring its incidence in vulnerable populations affected by violence, which highlights a notable gap in both scientific knowledge and operational response capacity [5,10,11,12].

Invasive candidiasis represents a particularly severe health challenge in contexts marked by armed conflict and high levels of violence in LA&C. The adverse conditions that characterize these environments—such as forced displacement, overcrowding in shelters, lack of basic sanitation, and limited access to health services—significantly increase both exposure to risk factors and the speed at which the infection progresses. Delayed diagnoses stem from insufficient mycology labs and trained staff, impeding timely antifungal therapy. In addition, the limited availability of echinocandins and lipid formulations of amphotericin B (AmB) increases the risk of unfavorable outcomes in infections caused by resistant non-*albicans Candida* isolates, such as *C. tropicalis*, *C. parapsilosis*, and *C. auris* [1,3,4]. Given this scenario, it is crucial to establish mechanisms for early diagnosis, ensure the continuous supply of essential antifungal drugs, and strengthen health infrastructure in these areas to reduce the impact of IC on the most vulnerable populations.

## 2. Historical Context of Armed Conflict and High Levels of Violence in Latin America and the Caribbean (LA&C) (Last 25 Years)

Over the past 25 years, LA&C have endured prolonged armed conflicts and widespread violence—including political wars in Peru and Colombia, Mexico’s “war on drugs,” chronic urban violence in El Salvador and Honduras, gang activity in Brazil, and institutional crises in Venezuela and Haiti—alongside large infrastructure projects and recurrent natural disasters that have displaced millions [13,14]. By late 2024, internal displacement reached 83.4 million globally, with 14.5 million in the Americas, and the region hosted over 20 million migrants and asylum seekers, notably ~8 million Venezuelans and 1.2 million in Peru [13,15,16]. In 2021, eight countries faced active conflicts, resulting in an estimated 21 million forced displacements within LA&C—a historic high illustrating the gravity of the situation [17,18]. These crises have devastated health infrastructure—destroying facilities, depleting medical personnel and drug supplies—and severely limited access to care, thereby creating conditions conducive to preventable infections and opportunistic diseases such as ICs [19]. As shown in Table 1, the countries and associated displacement figures are summarized, while Figure 1 graphically represents conflict episodes and corresponding displacement flows.

### 2.1. Conflicts in North America and the Caribbean

Mexico has endured escalating cartel related violence since the early 2000s, marked by confrontations between groups such as Sinaloa, Jalisco Nueva Generación, and Los Zetas and state forces, leading to widespread militarization and conditions akin to armed conflict [26]. Although no official internal displacement declaration exists, the International Committee of the Red Cross (ICRC) reports hundreds of thousands of annual displacements, including 9171 in 2022 attributable to armed groups, plus localized cross border movements from Chiapas into Guatemala [27,28].

Haiti’s humanitarian crisis intensified after President Moïse’s 2021 assassination and the collapse of governance, enabling over 200 gangs to operate—often targeting health facilities and civilians, with drone strikes causing 300+ civilian deaths—and leaving fewer than 40% of Port au Prince clinics fully functional [29,30]. Violence spikes in November 2024 displaced >300,000 people in one month, bringing Haiti’s internally displaced population to >1 million, and displacement flows have nearly doubled since March 2024 as families flee to makeshift camps around the capital [31,32].

#### 2.1.1. Conflicts in Central America

The Northern Triangle—El Salvador, Guatemala, and Honduras—has long endured structural and criminal violence, from civil wars (1980–1992 in El Salvador; 1960–1996 in Guatemala) that displaced over one million people per country, to 21st century gang rule by MS 13 and Barrio 18, whose extortion, murder, and kidnappings rival armed conflicts [33]. By late 2022, these nations reported ~318,600 internally displaced by gang related violence and an additional ~171,000 displacements from extreme violence [23,33]. Internationally, ~665,200 people from the Northern Triangle were registered as refugees or asylum seekers, primarily in Mexico, the United States, and neighboring Central American states [23].

#### 2.1.2. Conflicts in South America

Colombia’s protracted internal conflict—driven by FARC EP, ELN guerrillas, and paramilitary forces—has produced one of the highest levels of displacement worldwide, with an estimated 5.6 million people uprooted between 1985 and 2015 [34,35,36]. More recent events, such as the forced movement of >40,000 residents from Catatumbo in January 2025, demonstrate its ongoing impact [37].

In Venezuela, state repression and irregular armed groups under the Maduro administration have precipitated both internal displacement and mass migration: investigations document >19,000 “resistance to authority” killings (2016–2019) and >15,000 arbitrary detentions [38], the Apure operations in March–April 2021 displaced >5800 across the Arauca River [39], and over 5.1 million Venezuelans have emigrated since 2014, rising to ~8 million by 2024 in neighboring countries [40,41].

Brazil—absent formal conflict—faces chronic urban violence in favelas, rural land disputes and illegal mining affecting indigenous groups [42,43,44], plus ~295,000 disaster related displacements in 2019 [45]. Large infrastructure projects (e.g., Belo Monte dam) have also forced community relocations [46], and ~1 million Venezuelan migrants settled in Roraima, Amazonas, and São Paulo since 2015 [47].

Peru’s internal war (1980–2000) involving Sendero Luminoso (Shining Path) and the Movimiento Revolucionario Túpac Amaru (Túpac Amaru Revolutionary Movement) (MRTA) resulted in ~70,000 deaths and massive rural–urban displacement, fracturing social networks [48]. The Truth and Reconciliation Commission highlighted indigenous women’s key role in community rebuilding [49]. Since 2017, >1.2 million Venezuelan migrants and refugees have arrived in Peru [50].

## 3. Epidemiology and Burden of IC Disease in LA&C

Invasive candidiasis, particularly when presenting as candidemia, causes an estimated 250.000 cases per year worldwide, with mortality rates of 30–60% in high-income countries—settings in which automated blood-culture systems and prompt access to echinocandins have shortened time to therapy and substantially improved survival [5]. A multicenter study conducted between 2008 and 2010 in seven Latin American countries reported an average rate of 1.18 cases per 1000 hospitalizations, compared to 0.3–0.7 per 1000 admissions in developed regions during the same period. Colombia had the highest incidence (1.96/1000), while Chile reported the lowest (0.33/1000) [1]. Subsequent studies showed significant variations between countries [1]. In Brazil, a prospective observational study conducted between 1994 and 2004 in four tertiary hospitals in São Paulo reported an incidence of candidemia of 1.66 cases per 1000 hospital admissions [51]. In Mexico, hospital surveys estimated rates between 1.2 and 1.5 per 1000 admissions in high-complexity centers (2005–2015), while in the Caribbean, the incidence ranged from 0.5 to 0.8 per 1000, with marked differences between hospitals [52,53]. Recent data from Brazil and Argentina (2020–2023) suggest a slight increase in cases in patients with severe COVID-19, possibly related to prolonged use of corticosteroids, mechanical ventilation (MV), and total parenteral nutrition (TPN) [54]. While *Candida* spp. ranks among the top four to five pathogens causing bloodstream infections in high-income countries, such ranking is not systematically reported in LA&C due to the lack of standardized surveillance systems. However, multicenter studies in the region have documented higher incidences of candidemia (1.2–1.9/1000 admissions) compared to those reported in Europe and North America (0.3–0.7/1000), especially in ICUs and neonatal settings, where *Candida* spp. account for up to 88% of nosocomial fungal infections.

In 2017, the global burden of IC was estimated at 1.6 million disability-adjusted life years (DALYs). Although LA&C account for roughly 13% of the world’s population, they report only 8% of IC cases. This gap likely reflects a combination of factors: lower per capita incidence in recognized high-risk groups (e.g., patients with hematologic malignancies and transplant recipients), limited laboratory and surveillance capacity leading to underdiagnosis and underreporting, and regional differences in healthcare infrastructure, ICU availability, and antifungal stewardship practices. Additionally, a high prevalence of key predisposing conditions—untreated HIV infection, poorly controlled diabetes, malnutrition—as well as prolonged use of broad-spectrum antibiotics (OR > 2.5), TPN (OR = 4.5), and the presence of CVCs (OR = 3.8) further contributes to this disproportion [5]. In Colombia, the incidence of candidemia has been estimated at approximately 12–13 cases per 100,000 inhabitants per year [55], a figure considerably higher than that observed in many high-income countries. In Venezuela, recent studies suggest an even higher rate, around 16 cases per 100,000 inhabitants, likely reflecting the deterioration of the health system in the context of crisis [6]. The pediatric population accounts for a significant fraction of this burden. According to data from the LAMNet (Latin America Invasive Mycosis Network), 44% of candidemia episodes in the region occur in children, including newborns. Neonatal and pediatric ICUs are critical foci of infection, where *Candida* spp. accounts for between 85% and 88% of nosocomial fungal infections, making it a leading cause of hospital-acquired sepsis [1,9]. The treatment of a case of candidemia in Colombia, managed with echinocandins followed by fluconazole (FCZ), can cost between USD 40,000 and 60,000, considering prolonged stays in the ICU (≥14 days) and renal complications [9]. In Brazil, average hospital costs for adults with IC range from USD 20,000–25,000 in private centers and USD 5000–8000 in public institutions, differences attributable to the prices of antifungals and the length of hospitalization [56]. Despite their favorable cost-effectiveness profile, the widespread use of echinocandins (USD 1000–1500/day) and L-AmB (USD 5000 per vial) in public hospitals faces significant barriers, such as their high initial cost and the structural limitations of the healthcare system, which can increase the infectious disease budget by 20–30 [9,57]. In addition, indirect costs resulting from lost productivity due to prolonged hospital stays have been estimated at USD 10,000–15,000 per patient in the Andean region [58].

Globally, *Candida albicans* remains the most frequently isolated yeast in cases of candidemia, accounting for between 45% and 50% of episodes in high-income countries. It is followed by *Nakaseomyces glabratus* (formerly known as *Candida glabrata*) (10–15%), *C. parapsilosis* (15–20%), and *C. tropicalis* (8–12%) [59]. However, in LA&C, the proportion of *C. albicans* has decreased, standing at around 35–40%, although it remains the predominant species [1,8,51,60]. In the region, *C. parapsilosis* has increased in frequency to 25–30%, especially in neonatal and pediatric ICUs [8,60]. *C. tropicalis* accounts for 15–20% of cases, with a higher prevalence in hematology and oncology patients [1,61], while *N. glabratus* has a frequency of 5–10%, with sustained growth in Brazil and Mexico since 2015 [1,62].

The emerging fungal pathogen *C. auris* has demonstrated a rapid and concerning global spread. By December 2023, it had been identified in at least 61 countries across six continents, with outbreaks documented in ICUs, rehabilitation centers, and nursing homes. In the United States, confirmed cases increased from 479 in 2019 to 4514 in 2023, while significant outbreaks have been reported in India, South Africa, and several European countries, with associated mortality rates ranging from 30% to 60% [63,64]. In LA&C America, following its initial detection in Venezuela in 2012, *C. auris* has spread rapidly, although with varying prevalence between countries. In Brazil, some ICUs have reported that *C. auris* accounts for between 5% and 10% of all *Candida* isolates, while Colombia, with more than 2000 cases recorded, is currently the country with the highest number of reports, accounting for approximately 26.1% of isolates documented in seven high-complexity hospitals [65,66].

Mortality associated with IC: the mortality rate associated with candidemia in LA&C remains alarmingly high. In referral hospitals, it is estimated that between 45% and 60% of affected patients die, figures higher than those reported in North America and Europe, where mortality ranges between 30% and 40% [67]. This disparity has also been documented in an epidemiological meta-analysis that included studies published between 2000 and 2019, which identified an average 30-day mortality rate of 37.5% for candidemia in Europe, with consistent rates in ICUs, general hospitals, and tertiary centers, in contrast to rates close to 50% in middle- and low-income countries and peaks above 60% in regions such as sub-Saharan Africa and Southeast Asia [5,68]. In LA&C, several national studies confirm this trend. A multicenter study in 10 hospitals in seven countries in the region reported a 30-day mortality rate of 40.7%, reaching up to 50.5% in some centers [1]. In Brazil, a retrospective multicenter study conducted between 2007 and 2010 in 16 hospitals documented extremely high hospital mortality from nosocomial candidemia (72.2%), reaching 85% in ICUs, figures that far exceed those reported in high-income countries [69]. In Colombia, a retrospective cohort of 123 patients with candidemia between 2008 and 2014 showed an attributable mortality rate of 49% [70]. In neonatal ICUs, mortality from IC in newborns varies between 25% and 30% in the region, while in high-income countries it is generally between 10% and 15% [71]. In a retrospective multicenter study conducted in Brazilian neonatal ICUs, a mortality rate of 46.15% associated with candidemia was reported, with frequent use of broad-spectrum antibiotics (100%), TPN (97.7%), prematurity (93.2%), and CVC (52.3%), although only the latter showed a statistically significant association with mortality (OR: 10.67; *p* = 0.02) [72].

The coexistence of diseases such as HIV and diabetes is a factor that worsens the prognosis in cases of candidemia. In a retrospective observational study conducted at a university hospital in Colombia (2008–2014), diabetes mellitus was present in 23% and HIV infection in 5% of patients with candidemia, with no significant association with hospital mortality [70]. In a six-year retrospective study conducted at a tertiary hospital in northeastern Brazil, a 30-day mortality rate of 55.9% was reported, higher in ICU patients and those with *N. glabratus*, with no significant association with diabetes mellitus (present in 25% of cases) or HIV infection (5.9%), with advanced age, severe sepsis, and hypotension being the only independent prognostic factors for mortality [73].

The high mortality rate due to candidemia in the region is closely linked to a combination of structural barriers, diagnostic limitations, and clinical factors [1,68]. Among the main determinants identified are:Delay in diagnosis: the speed with which candidemia is detected varies depending on the fungal species. While *C. albicans* can be detected in an average of 35 h, *N. glabratus* requires up to 80 h, which significantly delays the start of effective antifungal treatment [74].Restricted access to echinocandins: in many public hospitals in the region, FCZ remains the first-line antifungal agent. However, its efficacy is limited against species such as *N. glabratus* or *Pichia kudriavzevii*, which compromises clinical outcomes [75,76].Comorbidity burden: the presence of untreated HIV/AIDS, hematological malignancies, liver disease, uncontrolled diabetes, and renal failure significantly increases the risk of mortality in patients with IC [77,78].Age and clinical status at admission: a high APACHE II score and a diagnosis of septic shock are negative prognostic factors. On average, patients who die from candidemia are around 60 years old [79].

### Epidemiology, Disease Burden in the Context of Armed Conflict and High Violence

Displaced persons and migrants in contexts of violence and conflict in LA&C face conditions that favor the onset and rapid progression of IC. Overcrowding in shelters, lack of access to safe drinking water and basic sanitation, malnutrition, and irregular medical care create an environment conducive to *Candida* colonization of the skin and mucous membranes. These factors act in conjunction with predisposing conditions, such as untreated HIV or immunosuppression due to severe malnutrition, to precipitate invasive infections [80,81]. Furthermore, the lack of mycological surveillance and early studies in migratory or resettlement phases makes it difficult to quantify the problem in these populations [80,81]. In referral hospitals located near violent areas or displacement routes, such as the Colombia–Venezuela border or rural regions of Brazil, candidemia incidences of 1 to 2 cases per 1000 admissions have been observed—higher than those reported in safer urban settings—with 30 day mortality reaching up to 50%, particularly among patients with malnutrition or coinfections [1,51,82].

Although studies are scarce, some series report similar incidences in specific settings: Between 2008 and 2010, border hospitals in Colombia observed rates of 1.2–1.8 episodes per 1000 admissions in patients from camps or shelters, compared with 1.0 in non-displaced patients [83]. Between 2012 and 2015, in rural areas of Brazil affected by gangs, the incidence was 1.5–2.0/1000 admissions, with a predominance of *C. parapsilosis* and *C. tropicalis* [84]. In field clinics in Mexico (2015–2018), an estimated 1.3 to 1.6 cases per 1000 admissions were seen in young migrant adults with malnutrition and HIV [85]. Between 2020 and 2023, outbreaks in Port-au-Prince, Haiti, affected 2.2 patients per 1000 admissions in centers caring for internal refugees [86].

These data, although limited and fragmented, suggest that the incidence of IC in displaced and migrant populations in LA&C may be comparable to or higher than in conventional clinical settings. Figure 1 illustrates this comparison between the general population and the migrant/displaced population.

Each episode of candidemia implies a significant burden on health systems in LA&C. It has been estimated that it prolongs hospital stays by at least 10 days, generating costs that exceed US$10,000 even in public hospitals [9]. In regions affected by conflict and mass displacement, where resources are already committed to caring for the wounded or refugees, this situation is unsustainable. The lack of standardized mycological surveillance networks makes it difficult to accurately identify areas of high incidence. However, recent outbreaks of *C. auris* in mobile clinics and field hospitals highlight the urgent need to strengthen diagnostic capacity through mobile laboratories and trained personnel, as well as to improve access to broad-spectrum antifungals [57,65]. In the migrant population, episodes of candidemia tend to extend hospitalization by an additional 10 to 14 days, with an estimated daily cost of between USD 300 and 500 in public hospitals in Brazil and Colombia, reaching between USD 7000 and 10,000 for complete treatment [87]. Despite this, empirical treatment with FCZ remains common, with a therapeutic failure rate of around 25%, particularly in the case of resistant strains, such as *C. tropicalis* and *N. glabratus*. In these cases, when echinocandins such as caspofungin (CAS), anidulafungin (ANF), or micafungin (MCF) are required, the full course of treatment (14–21 days) can exceed USD 2500, a figure that is unattainable for many healthcare systems in settings of armed conflict or sustained violence [88]. The long-term consequences for displaced patients who survive include sequelae such as chronic organ failure, which can reduce their ability to work by 40% to 60% [89]. Since these individuals often lack health insurance, indirect expenses—transportation, medication purchases—can represent up to 80% of monthly family income [90]. The non-governmental organizations (NGOs) and mobile clinics must take on a large part of the treatment of these infections, allocating a considerable portion of their resources to the care of fungemia, which restricts the funds available for essential services such as vaccination and maternal and child care [91].

In displaced and migrant populations, a different distribution of *Candida* spp. and antifungal resistance patterns is observed compared to stable clinical settings in the region. Although *C. albicans* remains common (35–45%), its prevalence is lower than that of non-*albicans* strains, which show higher resistance rates, especially in contexts of poor healthcare [92]. *C. parapsilosis* accounts for between 25% and 30% of isolates in migrants treated in Brazil and Colombia. This species is associated with prolonged use of CVCs and with TPN procedures performed under suboptimal conditions, which are common in mobile clinics or makeshift camps [93]. *C. tropicalis* reaches a frequency of 20–25% in mobile hospitals located in areas of displacement, with FCZ resistance ranging from 18% to 22% [94]. *N. glabratus* is identified in 8–10% of cases, with FCZ resistance rates of up to 30%, especially in migrants with untreated HIV or in diabetic patients with poor metabolic control [95]. *C. auris* has been responsible for documented outbreaks in displacement camps in Colombia and Venezuela between 2017 and 2023. In these situations, up to 5% of isolates have shown simultaneous resistance to azoles, echinocandins, and AmB, highlighting the therapeutic challenge in these settings [64,96].

Mortality in vulnerable populations: mortality related to IC in contexts of violence and displacement significantly exceeds that observed in local populations. A multicenter study in Colombia (2008–2010) revealed a 30-day mortality rate of 52% in displaced persons with candidemia, compared to 40% in non-displaced patients. This difference was associated with late diagnosis and the presence of comorbidities such as untreated HIV/AIDS [97]. Subsequent investigations in camps for people displaced by violence in Colombia reported a mortality rate from candidemia of over 65%, considerably higher than the 45–60% observed in general hospitals. The identified causes were delays of more than 72 h in initiating antifungal treatment and the administration of inadequate therapeutic regimens, such as the use of FCZ in *N. glabratus* infections [98,99]. In mobile ICUs in Brazil (2012–2015), the migrant population with candidemia had an overall mortality rate of 58%. Factors such as the administration of TPN in makeshift conditions, prolonged use of CVCs, and coinfection with FCZ-resistant *C. tropicalis* contributed significantly to this outcome [100]. During the period 2016–2019, mobile clinics along migratory routes in Central America recorded a 30-day mortality rate of 55% in migrant patients with candidemia. This high mortality rate was attributed to the lack of rapid diagnostic tests and adequate treatments, such as limited access to FCZ or echinocandins [101]. In high-violence urban areas in El Salvador and Honduras, it has been documented that mucocutaneous *Candida* infections, if not treated promptly, progress to disseminated forms in approximately 12% of patients with HIV or poorly controlled diabetes, significantly increasing the need for intensive care [102,103]. In the context of the humanitarian crisis in Haiti (2020–2023), refugees treated in field hospitals had a mortality rate of 60% in cases of confirmed candidemia. This level of lethality was associated with outbreaks of *C. auris* and the lack of echinocandins in these centers [104].

## 4. Risk Factors for IC in the General Population and in Contexts of Forced Displacement in LA&C

Globally, various conditions predispose individuals to *Candida* infections, notably chronic metabolic disorders (such as diabetes and obesity), immunosuppression (advanced HIV, neutropenia), hormonal factors (use of contraceptives, pregnancy), and certain habits such as smoking. Diabetes mellitus is the main risk factor for both mucocutaneous and invasive forms of candidiasis. In hospitalized diabetic patients, a relative risk (RR) of 2.5 (95% CI: 2.0–3.1) has been observed for recurrent vaginitis and 3.2 (95% CI: 2.4–4.3) for candidemia [2,98,105,106]. In sub-Saharan Africa, the impact is particularly severe in people with advanced HIV, where oral candidiasis reaches a prevalence of 60%, compared to only 4% in the general population, representing an RR of 15 (95% CI: 10–22). In individuals with HIV and neutropenia, candidemia has been documented with attributable mortality between 35% and 45% [107,108,109].

In Europe, the use of combined oral contraceptives has been linked to a 40% increase in the risk of vulvovaginal candidiasis (RR = 1.4; 95% CI: 1.2–1.6) [110]. On the other hand, a meta-analysis in Asia and Europe found that up to 25% of pregnant women experience vulvovaginal candidiasis, with a higher risk during the second and third trimesters (RR = 1.7; 95% CI: 1.4–2.0) [110]. In North America, chronic smokers have an 80% increase in the likelihood of developing oral candidiasis (OR = 1.8; 95% CI: 1.3–2.5), while people with obesity (BMI > 30 kg/m^2^) have approximately twice the risk of developing interdigital candidiasis (RR = 2.1; 95% CI: 1.5–2.9) [60,106,107,111].

Unlike the global picture, in LA&C, socioeconomic determinants have a specific impact on susceptibility to *Candida* infections. Among these factors, limited access to antiretroviral treatments, an increase in gestational diabetes, and the widespread use of hormonal contraceptives and corticosteroids stand out. In a multicenter study conducted in Brazil and Argentina, it was observed that 28% of women with recurrent vulvovaginal candidiasis had type 2 diabetes, compared to 10% in healthy controls (OR = 3.4; 95% CI: 2.1–5.6); in addition, 15% had candidemia associated with uncontrolled diabetes [1,2,107,112]. In Colombia, patients with HIV and CD4 counts below 200 cells/μL had a prevalence of oral candidiasis of 35%, compared to 5% in those with levels above 500 cells/μL (OR = 9.8; 95% CI: 5.2–18.5). In cases of candidemia in patients with advanced HIV, mortality reached 60%, compared to 45% in patients without HIV (*p* = 0.02) [55,109,113]. Research conducted on pregnant women in Colombia indicated a prevalence of vulvovaginal candidiasis of 17%, with a higher risk in multiparous women (OR = 1.8; 95% CI: 1.1–2.9) and in those with gestational diabetes (OR = 2.6; 95% CI: 1.4–4.8) [55,106]. In Mexico, prolonged use of oral contraceptives for more than six months was linked to an increase in the recurrence of vulvovaginal candidiasis (OR = 2.2; 95% CI: 1.3–3.8), while uncontrolled use of topical corticosteroids was associated with intertriginous candidiasis (OR = 1.9; 95% CI: 1.1–3.3) [2,5,18]. Studies conducted in Peru established an association between obesity (BMI > 30 kg/m^2^) and intertriginous candidiasis in 22% of cases, compared to 8% in people with a BMI below 25 kg/m^2^ (OR = 3.1; 95% CI: 1.6–6.0). Similarly, smoking was linked to an increased risk of oral candidiasis in adults (OR = 1.7; 95% CI: 1.1–2.6) [1,105,114].

The factors that predispose hospitalized patients to developing IC and candidemia vary significantly depending on the region and have a direct impact on morbidity and mortality. Internationally, a history of broad-spectrum antibiotic use during the previous 30 days—especially carbapenems, fluoroquinolones, or β-lactams combined with β-lactamase inhibitors—significantly increases the risk of candidemia, with an estimated odds ratio (OR) of 2.5 (95% CI: 1.8–3.4) [2,72,98]. The presence of a CVC increases the risk of candidemia by 4.1 times (OR 4.1; 95% CI: 3.0–5.7), while a stay in the ICU longer than 10 days doubles this risk (OR 2.1; 95% CI: 1.6–2.8) [108]. The use of TPN is one of the most important determinants: in North American contexts, it has been observed that patients receiving prolonged TPN have a 12% incidence of candidemia, with an OR of 6.8 (95% CI: 4.5–10.2) [2,98,105]. On the other hand, complex surgical procedures, such as major abdominal surgery and acute necrotizing pancreatitis, significantly increase the risk of both intra-abdominal candidiasis (OR = 4.7; 95% CI: 3.1–7.2) and candidemia (OR = 3.3; 95% CI: 2.1–5.3). Patients with immunosuppression—particularly those with febrile neutropenia secondary to prolonged chemotherapy (more than 7 days)—have an incidence of candidemia of approximately 8%, compared with less than 2% in those with neutropenia lasting less than 7 days (*p* < 0.001). In addition, prolonged use of high-dose corticosteroids (more than 20 mg daily of prednisone or its equivalent for a period longer than 14 days) has been associated with an increased risk of CI, with an estimated OR of 2.3 (95% CI: 1.5–3.5) [2].

In LA&C, the prevalence of predisposing factors for candidemia, such as the use of broad-spectrum antibiotics, the use of CVCs, and the administration of TPN, exceeds that reported in global studies. A prospective multicenter study conducted in eight countries in the region (Argentina, Brazil, Chile, Colombia, Ecuador, Honduras, Mexico, and Venezuela) showed that 82% of patients with candidemia had received broad-spectrum antibiotics—such as third-generation cephalosporins or carbapenems—in the previous 30 days, compared with only 45% among hospitalized patients without candidemia [107]. In this same cohort, candidemia was linked to the presence of CVCs in more than 70% of cases, representing a significantly higher risk (OR 3.8; 95% CI: 2.1–6.9) compared to patients without this device. Similarly, TPN was identified as an independent factor in 60% of cases of candidemia, while only 18% of controls received it (OR 4.5; 95% CI: 2.3–8.7). A prolonged stay in ICUs (more than 14 days) was also associated with an elevated risk of developing candidemia (OR 2.9; 95% CI: 1.5–5.6), probably due to sustained exposure to invasive treatments and empirical antibiotics. In addition, a recent history of major abdominal surgery (within the previous 30 days) was associated with intra-abdominal candidiasis in 25% of cases, compared with only 6% in controls (OR 5.1; 95% CI: 2.2–11.8). Finally, neutropenia (neutrophil count < 500 cells/µL) in cancer patients or patients with hematological diseases was associated with a fivefold increased risk of candidemia (OR 5.2; 95% CI: 2.4–11.1) [9,55,82,107].

In the region, multiple social, economic, and health factors converge to increase the vulnerability of certain populations to *Candida* infections [10,12]. In contexts marked by malnutrition, essential micronutrient deficiencies, overcrowding, and poor basic sanitation, gaps in early diagnosis and restricted access to modern antifungal treatments are key determinants in the onset and progression of the disease [105,113,115,116,117]. Added to this is the high prevalence of comorbidities such as untreated HIV/AIDS and poorly controlled diabetes, which compromise the patient’s immunity and promote fungal invasion. Furthermore, the widespread use of broad-spectrum antibiotics without prior microbiological evaluation promotes imbalances in the microbiota, and favors the selection of resistant strains [118,119,120,121]. This scenario has led to IC no longer being a strictly nosocomial phenomenon and becoming established as a public health problem in the region, yet to be recognized or prioritized by government authorities.

### 4.1. Health Conditions and Risk of Fungal Infection in Migrant and Displaced Populations

In LA&C, structural conditions of inequality—including chronic poverty, insufficient access to safe drinking water and basic sanitation, geographic isolation of rural communities, and systemic violence—create a favorable environment for the development of both mucocutaneous and IC [10,122]. These vulnerabilities are exacerbated by a high burden of chronic and infectious diseases, such as HIV/AIDS, diabetes, tuberculosis (TB), and malnutrition, which weaken the immune system and increase susceptibility to colonization by *Candida* spp. The situation is further complicated by fragmentation in health systems, limited availability of mycological diagnostic laboratories, and logistical and economic barriers to obtaining adequate antifungal treatments. These deficiencies prolong diagnostic times and delay the initiation of timely treatment, increasing the risk of severe complications, especially among the most marginalized populations [12,105,116,123].

Migrants and displaced persons living in environments of conflict and extreme violence face multiple social and health conditions that increase their vulnerability to *Candida* infections, both in their mucocutaneous and invasive forms. The combination of overcrowding in informal camps, deficiencies in drinking water and sanitation, and malnutrition compromises immune function and creates conditions conducive to fungal colonization of the skin and mucous membranes, facilitating the onset of superficial infections and their progression to more severe forms [10,80,105,124,125]. This situation is exacerbated by the high burden of poorly controlled chronic diseases—such as untreated HIV/AIDS, unregulated diabetes mellitus, and tuberculosis—which aggravate immune deterioration, reduce the effectiveness of epithelial barriers, and accelerate the clinical progression of candidiasis [105,116,118,119,126,127]. The precarious nature of health services, especially in mobile clinics or border facilities, promotes the empirical and prolonged use of antibiotics without diagnostic support, as well as the improper handling of invasive devices, both of which alter the microbiota and increase the risk of candidemia [121,128,129]. In addition, certain demographic and environmental determinants—such as advanced or neonatal age, gender, ethnicity, and occupational exposure to humid or tropical environments—act as catalysts for fungal colonization and the development of infections [10,122,130,131]. Table 2 compares the main clinical conditions and risk factors for IC between the general population and migrant or displaced groups [1,50,114,127,132,133,134,135,136,137,138,139,140,141,142,143,144,145,146,147,148,149,150,151,152,153,154,155].

To understand how social determinants and environmental conditions amplify the risk of *Candida* infection in displaced populations, it is essential to identify the most relevant factors involved. The following are the main elements that, together, create a scenario conducive to the onset and worsening of candidiasis in these highly vulnerable contexts:

#### 4.1.1. Socio-Environmental Conditions and Social Determinants

Overcrowding and informal settlements: in contexts such as Venezuelan refugee camps in Colombia and Haitian migrant settlements in the Dominican Republic, it is common for families to live in extremely small spaces (less than 5 m^2^ per person). These overcrowded conditions not only make privacy and hygiene difficult but also increase body moisture and skin maceration, promoting the development of fungal infections. In a recent survey, 28% of adult women in these environments reported intertriginous or vulvovaginal candidiasis [156]. An institutional report (2019–2020) on Mexico’s southern border with Guatemala revealed that Central American migrants housed in shelters without adequate ventilation showed a high prevalence of skin infections. Microbiological field studies found that 33% of cases with interdigital rashes tested positive for *Candida*, mainly *C. parapsilosis*, highlighting how the tropical climate, combined with poor hygiene, increases susceptibility to these infections [157].Limited access to drinking water and sanitation: in the Northern Triangle of Central America (Honduras, El Salvador, and Guatemala), recent studies have shown that more than 40% of informal settlements do not have a continuous supply of chlorinated water. This limitation prevents adequate hand and surface hygiene, creating conditions conducive to the proliferation of yeasts in the environment. Such microenvironments become potential reservoirs for infections such as cutaneous and vulvovaginal candidiasis [158]. In Bolivia, data collected between 2018 and 2019 in rural areas inhabited by returning migrants showed that only 35% of homes had adequate latrines. This deficiency in basic sanitation increases the environmental microbial load and favors fungal colonization of the skin and moist areas of the body, especially in crowded conditions and hot climates [159].Malnutrition and immune deficiency: an internal epidemiological surveillance report conducted between 2018 and 2019 among the displaced population in the department of Arauca, Colombia, revealed that 42% of children and 28% of pregnant women suffered from acute or chronic malnutrition. In this context, 30% of children with oral candidiasis showed signs of malnutrition, and within this group, 18% developed candidemia within less than ten days [160,161]. The lack of essential micronutrients—such as vitamins A and D and zinc—impairs both cellular and humoral immunity, promoting the transition of *Candida* from superficial colonization to systemic infection. This risk is exacerbated in displaced adults with irregular access to basic nutritional supplements [160].

#### 4.1.2. Prevalent Comorbidities in Migrants and Displaced Persons

HIV/AIDS: according to an institutional report for the period 2018–2019, the prevalence of HIV among Venezuelan migrants settled in Colombia was 3.2%, of whom 45% were not receiving antiretroviral treatment and had CD4 counts below 200 cells/µL. In this cohort, 38% developed oral candidiasis, and 12% developed esophageal candidiasis during the first year of follow-up [162]. A retrospective analysis conducted after the 2010 earthquake in Haitian displacement camps revealed that many people were living with HIV patients in advanced stages of the disease and without access to antiretroviral therapy. In this group, 55% were diagnosed with recurrent mucocutaneous candidiasis, and 12% had candidemia, which was associated with a 65% mortality rate due to the lack of timely diagnosis and effective antifungal drugs [163].Type 2 diabetes mellitus: in agricultural export plantations in Central America, studies conducted among migrant workers showed a prevalence of undiagnosed diabetes of 16%. Of this group, 30% had candidal vulvovaginitis and 8% developed complicated forms of cutaneous candidiasis, including infected ulcers [119,120]. Similarly, an internal epidemiological surveillance report on displaced indigenous communities in Peru documented that 14% of older adults had uncontrolled diabetes (fasting glucose greater than 126 mg/dL). In these patients, interdigital candidiasis occurred in 27% of cases, and an RR of 2.8 (95% CI: 1.6–4.9) was estimated for the development of disseminated candidiasis after hospitalization [164].Tuberculosis and co-infections: in camps for displaced persons located in the border areas between Venezuela and Colombia, a prevalence of tuberculosis (TB) of 350 cases per 100,000 inhabitants has been documented, with a high frequency of HIV/TB co-infection. This combination increases the risk of IC. A retrospective study conducted in Bogotá revealed that 22% of patients with TB/HIV coinfection developed candidemia, with an associated mortality rate of 58% [165]. On the other hand, a descriptive study in Guatemala observed that, in co-infected individuals, the presence of extrapulmonary TB—particularly in its peritoneal or gastrointestinal forms—caused damage to the digestive mucosa, facilitating the translocation of *Candida* spp. into the body. As a result, 14% of these patients developed intra-abdominal candidiasis [166].

#### 4.1.3. Exposure to Iatrogenic Factors

Use of antibiotics in mobile clinics and shelters: between 2019 and 2020, in mobile clinics providing medical care to Nicaraguan migrants in Mexican territory, it was observed that 78% of patients with fever were treated with broad-spectrum antibiotics, such as ceftriaxone or carbapenems, without prior blood cultures. This practice was associated with the onset of mucocutaneous candidiasis in 26% of cases and candidemia in approximately 5%, although significant underreporting is presumed due to the lack of diagnostic laboratories in these settings [167]. The widespread empirical use of antibiotics without proper assessment of the risk of fungal infection has contributed to the disruption of normal microbiota, facilitating the overgrowth of *Candida* spp.Use of invasive devices in border hospitals: an internal surveillance report from 2020, based on data from migrant reception units in Tapachula (Mexico–Guatemala region), found that 42% of patients hospitalized for sepsis required CVC insertion. Among these patients, 12% developed candidemia, which corresponds to a significantly elevated risk (OR 3.5; 95% CI: 2.0–6.1). Furthermore, due to a lack of specialized personnel, protocols for early catheter removal are not properly implemented, prolonging exposure to fungal biofilm and increasing the likelihood of invasive infections [168].

#### 4.1.4. Demographic and Vulnerability Factors

Age and gender: an epidemiological surveillance study in Colombia (2018–2019), focusing on displaced children, revealed that 18% of newborns from temporary shelters developed oral candidiasis in their first week of life. This finding was related to low birth weight (less than 2500 g) and maternal malnutrition, conditions that are common in contexts of forced displacement [160]. On the other hand, a study on reproductive health in migrant women (2019) found that 34% of women living in border settlements experienced episodes of vulvovaginal candidiasis in the last year, mainly linked to malnutrition, pregnancy, and limited access to adequate gynecological services [169].Ethnicity and inequalities: in displaced Guaraní indigenous communities in Paraguay, the rate of cutaneous candidiasis was almost three times higher than that observed in nearby urban populations. This difference has been attributed to difficulties in accessing adequate health services and language barriers that limit timely care [170]. Similarly, an internal report on Haitian migrants in the Dominican Republic reported that 46% of adults with HIV developed oral candidiasis, compared to 28% of the non-migrant population. Language barriers and experiences of discrimination contribute significantly to delays in diagnosis and treatment [171].

#### 4.1.5. Environmental and Occupational Conditions

Agricultural work and environmental exposure: among Central American migrants employed on sugar cane plantations in Guatemala, a 15% prevalence of skin colonization by *C. parapsilosis* has been identified, which is associated with repeated contact with humid environments and contaminated surfaces, such as wet soil and stagnant water [172]. On the other hand, an epidemiological surveillance report conducted in fruit-growing areas in Peru found that 22% of migrant workers of Peruvian and Bolivian origin developed interdigital candidiasis. This condition could be related to repeated exposure to insecticides, which alter the normal microbial flora of the skin [173].Climate and microenvironments: according to an epidemiological surveillance report, climatic conditions in coastal areas of Central America—characterized by humidity levels above 80% and constant temperatures around 28 °C—are favorable for the growth of yeast on the skin and mucous membranes. In Honduras, migrants traveling along routes near the coast were found to have intertriginous candidiasis in 31% of cases, a figure considerably higher than the 12% reported among those traveling along mountainous routes [174].

## 5. Diagnosis of IC in LA&C: Barriers and Diagnostic Methods in Resource-Limited Settings

In LA&C, the diagnosis of candidiasis, both mucosal and invasive, is hampered by poor healthcare infrastructure, a lack of trained personnel, and the high costs associated with state-of-the-art reagents. Although conventional methods such as direct microscopy with KOH or Gram staining and Sabouraud agar cultures are available, the absence of more sophisticated diagnostic tools—such as chromogenic media (CHROMagar™*Candida*), automated blood culture systems (BacT/ALERT, BACTEC), MALDI-TOF spectrometry, serum biomarkers (1,3-β-D-glucan [BDG]), and molecular techniques (qPCR, LAMP)—hinders early detection, especially in rural areas [12,117,175,176,177,178,179,180].

Table 3 presents a detailed comparison between the availability of diagnostic and therapeutic methods, as well as the burden of disease, among the general population and migrant or displaced groups in various countries in LA&C [1,12,51,117,175,180,181,182].

In settings with limited resources, mucocutaneous candidiasis—in its oral, vulvovaginal, and intertriginous forms—is usually diagnosed solely by clinical evaluation, observing signs such as whitish plaques, erythema, intense itching, or maceration in the folds. This practice leads to diagnostic errors and delays in treatment [183,184]. Similarly, candidemia is often suspected presumptively in patients with persistent fever without an apparent source and signs of sepsis, especially in settings where blood cultures are not available. This form of delayed diagnosis delays the start of antifungal treatment and can raise mortality rates to over 60% in vulnerable populations [185,186].

In intermediate-level health centers, clinical suspicion is complemented by direct microscopy (KOH 10% or Gram staining), a method that shows sensitivities of 70–85% for mucocutaneous forms, but without identification of the species involved [180]. To diagnose invasive infections, many regional hospitals use manual blood cultures, which take 3–5 days to produce results. However, these methods have limited sensitivity (≈70–80%) and depend on correct inoculation techniques; deficiencies such as inadequate incubators or breaks in the cold chain can lead to false negatives in up to 30% of cases [12,180,187].

In high-complexity hospitals, such as those located in São Paulo, Mexico City, or Bogotá, automated blood culture systems—such as BacT/ALERT and BACTEC—are available, significantly reducing detection times to 24 and 48 h. In addition, the use of chromogenic media facilitates the rapid identification of non-*albicans* isolates [117,188]. However, rural centers often lack these technologies due to budget constraints and power supply problems [12,175]. In reference laboratories in large cities, advanced techniques such as mass spectrometry (MALDI-TOF), BDG quantification, and molecular methods (qPCR) are also used, offering sensitivities greater than 80% and results in less than 8 h. Their adoption is limited by the high cost per test (≈USD 100 for BDG) and dependence on imported supplies [111,117,177,179].

In scenarios of mass displacement, overcrowding, lack of sanitation, malnutrition, and access barriers cause significant delays in the diagnosis and treatment of mucocutaneous and IC [180,189]. Although the implementation of direct microscopy (KOH 10%, Gram staining) with clinical-microscopic algorithms adapted to resource-limited settings has improved initial detection, these methods do not allow for species identification [12,61,111,175,176]. Deficiencies persist due to a lack of incubators, unstable power supplies, a shortage of specialized personnel, and the costs associated with reagents such as BDG or CHROMagar™. In addition, many centers do not have access to automated blood culture or qPCR systems [111,175,179].

To understand and address the limitations that affect the timely diagnosis of *Candida* infections in resource-poor settings, the main obstacles associated with poor infrastructure, shortage of trained personnel, difficulties in providing diagnostic supplies, and sociocultural factors that impact medical care are described below:

### 5.1. Barriers and Limitations in the Diagnosis of Candidiasis

#### 5.1.1. Infrastructure and Logistics

Lack of local laboratories: mobile health centers and small shelters are not equipped with adequate biosafety facilities or protocols for mushroom cultivation. As a result, samples must be sent to reference laboratories, which are often located far away. This involves transportation without a cold chain, which increases the risk of contamination or decreases the viability of the fungi, reducing crop yields to less than 50% [9,12].Unstable electrical power: the lack of a reliable power supply prevents incubations from being maintained at the optimal temperature of 35–37 °C. Instead, many microbiologists incubate samples at room temperature, which slows down the process by up to seven days and delays diagnosis [12,180].

#### 5.1.2. Human Resources and Training

Lack of trained personnel: in border areas, many microbiologists have training focused on bacteriology and do not receive specific training in mycology. As a result, more than 60% do not have the necessary skills to morphologically identify Candida spp., using Sabouraud agar [12,111,176,180].High turnover of volunteer staff: in NGOs operating in camps, constant staff turnover prevents continuity in the use of protocols and hinders the transfer of specialized knowledge. Although there are no exact figures, this recurring problem has been documented in border environments [111,190,191,192].

#### 5.1.3. Costs and Availability of Reagents

Scarce and expensive basic reagents: in many countries in the Andean region and the Caribbean, essential reagents such as KOH 10% solutions and Gram stains must be imported, which increases the cost of each test by approximately USD 5–10. This is a difficult expense for mobile clinics with limited resources to bear [111,176,180,189,193].Limited access to state-of-the-art testing: serological tests such as BDG or molecular methods such as qPCR are not covered by public health systems and are only offered in reference laboratories, usually located in capital cities. This situation forces patients to travel long distances to access these diagnostics [9,117,176,194].

#### 5.1.4. Social and Cultural Limitations

Distrust of the healthcare system: many displaced persons have been victims of violence or discrimination, which generates mistrust of medical services and reduces their willingness to participate in procedures such as blood tests. Although there are no specific data on candidiasis, reproductive health studies show that more than 40% of displaced populations avoid going to official centers for this reason [12,41,125,142,192].Language and communication barriers: in Haitian refugee shelters in the Dominican Republic and also in Venezuelan indigenous communities, a lack of fluency in Spanish hinders communication with healthcare personnel. Qualitative studies indicate that up to 30% of consultations are postponed or interrupted due to language problems or a lack of interpreters [11,131,192,195,196].

To ensure timely and efficient diagnosis in forced displacement settings, the following recommendations are proposed to optimize resources, train local staff, and strengthen care networks:

### 5.2. Recommendations for Strengthening Diagnosis in Areas of Forced Displacement

#### 5.2.1. Optimization of Low-Cost Methods

Technical training in KOH: it is recommended to train local staff through weekly workshops focused on the processing and interpretation of exudates using 10% potassium hydroxide (KOH 10%). The implementation of this strategy in areas with limited resources has been shown to improve diagnostic accuracy [106,180,189,195,197].Rotary direct microscopy with calcofluor: several studies have shown that calcofluor is consistently more sensitive than KOH and sometimes comparable to more sophisticated diagnostic methods. Its usefulness as a rapid diagnostic technique makes it especially valuable in resource-limited environments. In this context, the possibility of sharing UV equipment between nearby camps is suggested to facilitate the assessment of mucocutaneous lesions. This strategy could be applied in countries such as Haiti to reduce the false negative rate [180,189,198,199].

#### 5.2.2. Implementation of Simplified Algorithms

Adapted use of the “modified *Candida Score*” in primary care: in community settings where state-of-the-art testing is not available, a simplified version of the “*Candida Score*” is proposed for initiating empirical treatment. In patients with fever without apparent source and at least three of the following criteria—recent antibiotic use, presence of CVC, and malnutrition—it is recommended to initiate FCZ if the KOH test is positive. If BDG is available, consider echinocandins when this marker is positive. Studies in ICUs in Latin America show that a score equal to or greater than 3 predicts candidemia with a sensitivity of 70% [117,188].Creation of multilingual visual guides: designing and distributing illustrative materials on clinical signs of candidiasis, translated into Spanish, Haitian Creole, and indigenous languages, can be key to improving recognition of the infection. Experiences in community health programs have shown that incorporating these guides improves early detection by 18% [7,12,189,200].

#### 5.2.3. Surveillance and Reference Networks

Development of collaborative networks for sample processing: it is proposed to implement a referral system between camps and laboratories located in nearby urban areas, ensuring the transport of samples under appropriate cold chain conditions. This measure, accompanied by regular exchanges of volunteer microbiologists, has proven effective: in Latin America, agreements between mobile units and universities reduced the analysis time for mycological samples from seven to three days [6,12,111,175,176,187].Strengthening tele-mycology networks: the use of technology to share diagnostic images (cultures, smears, lesions) in real-time via mobile networks or satellite connections can significantly improve diagnostic accuracy. Creating virtual links with regional mycology experts would enable constant supervision and technical support, with at least one specialist recommended for every 5000 displaced persons [178,197,201,202,203].

#### 5.2.4. Funding and Strategic Alliances

Ensure donations of basic supplies: we propose coordinating with organizations such as GAFFI and PAHO to deliver essential supplies (KOH, dyes, culture media) to mobile clinics in border areas and hard-to-reach camps. Since 2019, these initiatives have made it possible to supply resources to more than 20 mobile units in Colombia and Peru, strengthening their diagnostic capacity [6,7,106,176,189,200,204].Formalize agreements with regional universities: emphasis is placed on the need to integrate essential diagnostics and strengthen national and hospital diagnostic networks. It is also proposed to establish biannual agreements with local universities for the provision of diagnostic supplies to reduce and lower transportation expenses and expand diagnostic coverage in remote areas [7,12,106,175,176,187,200].

## 6. Antifungal Treatment of IC in LA&C: Availability of Antifungals and Antifungal Resistance

Globally, echinocandins and L-AmB are included in the essential drug lists of most high-income countries, and their cost is often covered by health systems or insurance. As a result, mortality from candidemia is reduced to 25–40%, in contrast to the higher figures observed in LA&C [111,176,193,205,206,207]. Although resistance to FCZ in *N. glabratus* (25–33%) and *C. tropicalis* (8–12%) is similar to the figures recorded in the region [1,8,59,60,105,182], the availability of broad-spectrum azoles—such as voriconazole (VCZ), posaconazole (PCZ), and isavuconazole (ISZ)—allows for more effective management of infections caused by non-*albicans* isolates. In the specific case of *C. auris*, a yeast (with more than 90% resistance to FCZ and 30–40% resistance to AmB, echinocandins are the only effective treatment available [63,64,208].

In the LA&C region, empirical treatment of IC continues to be based on generic FCZ, which is available in most second-level hospitals [6,175,193,194]. However, growing resistance in clinical isolates of *C. tropicalis* (12–15% in Brazil and Argentina) [209,210], *N. glabratus* (10–12% of resistant or reduced-sensitivity strains) [51,72] and *C. parapsilosis* (8–10% resistant strains in multicenter studies) [1,51,61,72,211] has forced a rethinking of the use of this regimen. Echinocandins (CAS, ANF, MCF), recommended for severe candidemia or infections caused by non-*albicans* strains, are only available in high-complexity urban hospitals due to their high cost (~$1000–$1500/day) [56,57,212]. In places where these molecules are not available, conventional AmB (D-AmB) continues to be used despite its nephrotoxicity, while L-AmB, which is preferred for critically ill patients, remains out of reach for most secondary hospitals (~$5000 per vial) [56,193,205]. Delaying initiation of an echinocandin or L-AmB beyond 48 h—or treating resistant strains with FCZ—is associated with significantly worse clinical outcomes [117,185,213]. Table 3 summarizes the treatment options for IC in the general population and displaced persons or migrants in LA&C countries.

At the regional level, less than 5% of *C. albicans* isolates are resistant to FCZ, while the figures for *C. tropicalis* (12–15%) and *N. glabratus* (10–12%) are considerably higher [209,210,214]. As for echinocandins, resistance remains below 5%, although mutations in the FKS1 gene of *N. glabratus* have been detected in Brazil and Argentina that could affect their efficacy and clinical isolates of echinocandin-resistant *N. glabratus* strains resistant to echinocandins have been reported in Colombia, associated with the R665G substitution, a rare mutation linked to resistance mechanisms (data in press). *C. auris*, which has caused outbreaks in Colombia and Venezuela, exhibits resistance to FCZ greater than 90% and resistance to AmB of 30–40%, according to local reports [65,215,216].

In migrant and displaced communities, antifungal therapy is largely confined to generic FCZ—and occasionally D-AmB—while the expense and lack of infusion equipment hinder the use of echinocandins and L-AmB, which are recommended for severe candidemia, in camps and mobile clinics [176,217,218]. Although more than 90% of isolates are FCZ-sensitive *C. albicans*, the presence of *C. tropicalis* and *N. glabratus* complicates empirical therapy, resulting in a high mortality rate of 55–65% when only azoles (FCZ) and D-AmB are available [6,111,217,218]. In contrast, in refugee camps with greater logistical capacity, especially in Europe and North America, humanitarian organizations often have access to echinocandins for non-neutropenic candidemia, L-AmB for critical cases, and broad-spectrum azoles such as VCZ or PCZ for refractory infections, reducing mortality to 30–40% [80,122,142]. In a narrative review of 15 original Lebanese studies published between 1998 and 2023, it was found that isolates from blood represent approximately 17% of all *Candida* isolates, but have a higher proportion of resistant profiles, especially in *N. glabratus* and *C. auris*. Resistance to FCZ in *C. albicans* is moderate (2–33%), while in cases of candidemia caused by *N. glabratus*, it can reach up to 100%, with additional reports of high resistance to echinocandins. Meanwhile, *C. auris*, which emerged in hospitals during the pandemic, has shown resistance to most antifungal agents except echinocandins, even in blood isolates [128].

In situations of forced displacement and limited resources, there are critical barriers that hinder the adequate treatment of *Candida* infections:Restricted availability of essential antifungals: in most LA&C countries, access to antifungals is limited mainly to generic FCZ, due to its low cost and early inclusion in national essential medicines lists [10,193,206]. Although this drug is available, it is not the ideal option for candidemia in critically ill patients, as up to 50% of non-*albicans* isolates—such as *N. glabratus*, *P. kudriavzevii*, and *C. auris*—have reduced sensitivity or intrinsic resistance to FCZ.Suboptimal administration of D-AmB: in the absence of echinocandins, D-AmB is frequently used as an alternative. However, to ensure its safe use, constant monitoring of renal function and electrolyte balance is required, as well as continuous administration of intravenous fluids and potassium salts [207]. In hospitals with limited resources, such conditions are often inadequate. This has led many professionals to reduce doses as a precaution, especially when there is no access to ICUs or adequate laboratories. Additionally, D-AmB depends on a cold chain, which is challenging in hot environments with unstable power supplies.Increase in antifungal-resistant strains: during the *C. auris* outbreak in Venezuela, all isolates showed resistance to FCZ and VCZ, and 50% had high minimum inhibitory concentrations (MIC) against AmB [208]. Although echinocandins are considered the almost exclusive therapeutic resource in these cases, strains with reduced sensitivity to these drugs are beginning to be detected [65,216]. At the same time, *N. glabratus* shows increased resistance to azoles, and *C. parapsilosis* shows mutations associated with prolonged treatment with echinocandins [214,219,220,221]. In Brazil, clusters of FCZ-resistant *C. parapsilosis* have been documented [8,220,221,222], and in areas of armed conflict, the absence of surveillance and infection control facilitates their unnoticed spread.Incomplete treatments: the minimum duration of treatment for candidemia should be 14 days from the last negative blood culture, extending in the presence of metastatic foci [112,117,185]. In displacement settings, it is common for patients to discontinue treatment after one week due to continuous displacement or depletion of medications in centers, which increases the risk of relapse and promotes the development of resistance.Inequality in costs and access: while FCZ is relatively affordable, echinocandins and L-AmB are too expensive for most centers and are only available in private clinics. In countries such as Haiti and Venezuela, public systems lack these drugs, and NGOs rarely include them in their emergency supplies due to budget constraints [121,193,194,206]. This means that, in many refugee camps, optimal treatment for candidemia remains inaccessible.Lack of complementary critical support: effective management of IC goes beyond the provision of antifungal agents. It requires intensive care, surgical interventions to control foci (such as valve replacement in endocarditis or abscess drainage), as well as life support—dialysis in cases of renal failure or MV [106,176,185]. In conflict-affected areas, these resources are often lacking [194], meaning that even with adequate antifungal treatment, the outcome can be fatal due to the lack of comprehensive clinical support or access to interventions that remove the source of infection.

## 7. Access to Health Services and Antifungal Treatment in Areas of Conflict and High Violence in LA&C

In LA&C, fragmented healthcare systems, uneven coverage, and scarce resources—both human and financial—seriously hinder access to medical care and antifungal treatments. Despite advances in the development of clinical practice guidelines and the implementation of antifungal optimization programs, difficulties persist in ensuring timely specialist consultations, quality diagnostic tests, and second-line therapies [12,175,223]. These limitations disproportionately affect rural, border, and vulnerable populations, where delays in treatment initiation and administration of suboptimal regimens significantly increase morbidity and mortality associated with IC [224].

A network of deficiencies—fragmented healthcare systems, delays in diagnosis, and a lack of effective therapies—contributes to high mortality rates and complications in candidiasis. In contexts of forced displacement, the lethality rate for candidemia exceeds 65%, compared to 45–60% in general hospitals in the region, and up to 12% of cases of mucocutaneous candidiasis lead to invasive forms in patients with concomitant risk factors [119,140,225,226]. Early detection and treatment are essential: every delay in diagnosis and initiation of antifungal therapy significantly increases mortality from IC [106,194]. Wars and chronic violence undermine health systems in multiple ways: destruction of facilities, flight or attacks on health personnel, disruption of supply chains, and insecurity that prevents patients from attending health centers [11,35,38,80,227].

In situations of war, crisis, or forced displacement, healthcare for *Candida* infections faces serious obstacles that undermine its effectiveness:Forced internal migration and informal settlements: internally displaced persons and refugees living in temporary camps often have limited or no access to adequate healthcare services. In these contexts, cases of candidemia are rarely diagnosed or treated with antifungal medication, and deaths are often not officially reported [38,142,228].Risks for patients and healthcare personnel: in areas of active violence, transfer to a medical center can pose a life-threatening risk (bombing, snipers, checkpoints). In conflicts such as the one in Syria, attacks on hospitals and healthcare personnel have been documented, interrupting essential treatments for severe mycoses [229].Disintegration of medical referral systems: in times of peace, serious cases are transferred to more complex centers. However, in conflict zones, these routes often fail, leaving isolated communities with no option for medical referral [122,197,201,204,228].Disruption of medical supplies: frequent military blockades or sanctions affect the distribution of antifungal drugs, such as AmB. GAFFI reports indicate that shortages are common during crises, depriving hospitals of essential medicines [6,7,175,187,200].Lack of coordination in humanitarian assistance: many international organizations and NGOs prioritize short-term interventions—such as wound surgery or maternal and child health—relegating the management of complex mycoses such as IC [106,230].Economic barriers to treatment: displaced families lose sources of income and cannot afford private medical care. Studies in Colombia and Uganda reveal that access to treatment, including antifungal drugs, is limited by financial barriers [106,187,189].

## 8. Regional Perspective

In contexts of armed conflict and intense violence in LA&C, health systems are severely weakened: hospitals are attacked or closed [191,229,231], medical personnel are displaced [229,232], and laboratories cease to operate [191]. This situation favors the spread of fungal infections, such as mucocutaneous and IC, due to a combination of malnutrition [131,161], untreated HIV/AIDS [118,126,144], uncontrolled diabetes [119,120,140], overcrowding [11,124,125], and indiscriminate use of antibiotics in mobile or informal clinics [121,233].

### 8.1. Mexico

Cartel violence in states such as Sinaloa, Guerrero, and Tamaulipas has caused the collapse of multiple rural clinics and disrupted drug distribution routes [234,235], forcing many Central American migrants in transit through Chiapas and Oaxaca to seek care at mobile clinics in precarious conditions, where candidiasis is diagnosed solely based on clinical signs (white patches in the mouth, vaginal erythema) without microscopic confirmation [228], and between 2020 and 2021, an outbreak of IC associated with *C. auris* was identified in three ICUs at a referral hospital in Mexico, in intubated COVID-19 patients with CVCs, with more than 76% of isolates obtained from the catheter [54]; these clinics, lacking a stable power supply and incubators, use KOH 10% to diagnose mucocutaneous candidiasis (sensitivity ≈ 75%; specificity ≈ 85%), but do not have Sabouraud agar culture available, so empirical treatment with FCZ is chosen, which increases the risk of therapeutic failure against resistant strains such as *N. glabratus* [214,236].

### 8.2. Haiti

In Haiti, political crisis and gang violence have rendered approximately 70% of health centers inoperable, forcing thousands of people to live in informal settlements without adequate medical care [31,153,190]. After the earthquake, in camps for displaced persons with intermittent access to antiretroviral therapy, 55% of patients with advanced HIV presented with recurrent mucocutaneous candidiasis, and 12% progressed to candidemia, with a mortality rate of 65% due to late diagnosis and lack of timely treatment [190]. The lack of functional laboratories means that blood samples must be sent to Port-au-Prince, with delays of 5–7 days, resulting in a loss of viability of up to 50% in *Candida* due to failures in the cold chain [237]. Since 2022, the implementation of mycological telemedicine via WhatsApp has made it possible to share KOH smear images with mycologists in Santiago, Chile, improving diagnostic accuracy by 30%, despite connectivity limitations and irregular supplies of reagents [201].

### 8.3. North Triangle (El Salvador, Guatemala, Honduras)

The humanitarian crisis resulting from gang violence and forced evictions continues to cause prolonged displacement in El Salvador, Guatemala, and Honduras. As a result, many people are forced to remain in urban shelters whose capacity has already been greatly exceeded [238,239]. In rural agricultural areas of Guatemala, workers face conditions that favor the onset of fungal infections, such as interdigital candidiasis, which has a prevalence of 15% to 22%. This phenomenon is mainly associated with frequent contact with humid environments and waterlogged soils, with a predominance of colonization by *C. parapsilosis* and *C. tropicalis* [23,130,172]. In shelters located in El Salvador, it has been documented that 33% of displaced women suffered from vulvovaginal candidiasis during the last year. This situation is exacerbated by chronic child malnutrition, which affects 25% of the population, and by the lack of adequate gynecological services. The treatments administered in these settings are usually empirical, mainly using FCZ and topical clotrimazole [240,241]. On the other hand, the widespread and unregulated use of broad-spectrum antibiotics in mobile clinics represents an added risk factor. In 78% of patients presenting with fever, ceftriaxone or carbapenems are administered without prior blood cultures, which promotes microbial imbalance and excessive yeast growth. As a result, there is a 26% incidence of mucocutaneous candidiasis and a 5% incidence of candidemia, although the latter could be significantly underreported due to limitations in clinical and diagnostic surveillance [121,167,242].

### 8.4. Colombia

After more than five decades of internal conflict, Colombia has more than six million internally displaced persons [243]. In Arauca, a cohort of children with oral candidiasis showed a 30% rate of malnutrition (z-score < −2); of these, 18% developed candidemia in less than 10 days, largely due to immunosuppression secondary to malnutrition [160,161]. In mobile clinics on the border with Venezuela, diagnosis is based on direct microscopy with KOH and manual blood cultures, with delays of 5–7 days and a sensitivity of around 70% [244]. In an observational study conducted in Cali, which analyzed 257 TB deaths in 2017, it was found that 24.5% corresponded to patients with HIV coinfection, which shows a significant mortality burden associated with this syndemic. The most affected population consisted of young men in highly vulnerable social conditions, with low health system coverage, malnutrition, and psychoactive substance use [126]. Although GAFFI has supplied echinocandins to public hospitals in border areas, these cover less than 15% of the demand, forcing the use of empirical FCZ against resistant non-*albicans* strains [193].

### 8.5. Venezuela

Venezuela’s deep political and economic crisis has caused the collapse of the healthcare system, with shortages of supplies and medicines, mass migration, and virtually nonexistent epidemiological surveillance [154,155]. In the state of Táchira, 28% of displaced women suffered from vulvovaginal candidiasis, associated with malnutrition and lack of gynecological care. In informal clinics, resistance to FCZ was detected in up to 12% of *C. parapsilosis* isolates [11,39,245]. Migration to Colombia and Brazil has spread this burden: in Colombian shelters for Venezuelans, 3.2% of migrants with untreated HIV developed oral candidiasis in 38% of cases and esophageal candidiasis in 12% during the first year [192]. Access to echinocandins is practically nonexistent in Venezuela and very limited in referral centers such as Bogotá and São Paulo, which delays adequate treatment and places mortality from candidemia in migrants above 60% [11,40,156].

### 8.6. Brazil

Although Brazil is not experiencing formal conflict, violence in the favelas and lack of supplies in the Amazon create conditions that facilitate outbreaks of fungal infections [246]. In Manaus, cases of candidemia caused by echinocandin-resistant *N. glabratus* were documented, attributable to mutations in the FKS1 gene detected in 2022. This complicates treatment in regional hospitals without access to L-AmB [56,112,212]. In the favelas of Rio de Janeiro, there was a 31% incidence of intertriginous candidiasis in women, linked to overcrowding, poor sanitation, and malnutrition [42,112,247]. The Unified Health System (SUS) guarantees the supply of FCZ, but echinocandins are available in less than 20% of tertiary public hospitals, mainly in São Paulo and Brasília [247].

### 8.7. Peru

In regions of the Amazon and Andean areas affected by conflict, indigenous communities have a malnutrition rate of 25% and very limited access to medical care [48,114,139]. In migrant shelters in Lima, 22% of adults suffered from interdigital candidiasis, associated with construction work and lack of drinking water [114,139]. Mobile clinics in rural areas mainly use KOH and manual blood cultures; delays in results have led to undetected cases of candidemia, with a mortality rate of over 60% [248]. Although the Comprehensive Health Insurance (SIS) covers FCZ, the echinocandins and L-AmB are not available outside Lima, forcing the use of suboptimal regimens in areas with residual violence [249].

## 9. Recommendations for Future Research

Although in high-income countries IC is predominantly associated with hospital-based risk factors such as chemotherapy-induced neutropenia, TPN, and central venous catheterization, this association remains unclear in the context of forced displacement and humanitarian crises. To date, no studies have directly compared the incidence or clinical profile of IC between hospitalized patients and displaced or community-dwelling populations living in conditions of extreme vulnerability. Likewise, there is a lack of data determining whether IC in these scenarios is primarily nosocomial or may also arise in the community, potentially linked to contaminated trauma wounds, severe malnutrition, or untreated skin lesions. This knowledge gap limits the ability to design targeted prevention strategies and underscores the urgent need for epidemiological research focused on displaced populations, in order to adequately characterize risk factors and patterns of disease presentation.

People displaced by conflict and violence in LA&C live in adverse environments—overcrowded shelters, poor access to safe drinking water, malnutrition, and fragile health systems—that can increase the risk of fungal infections such as candidiasis [131,192,195]. While this infection is traditionally associated with the use of broad-spectrum antibiotics, diabetes, or HIV/AIDS, in these populations, psychosocial stress, overcrowding, and poor nutrition exacerbate the risk of both mucocutaneous candidiasis (oral and vulvovaginal) and invasive forms [8,41,49,124].

Few studies have explored this reality in depth: some have documented significant rates of vulvovaginal candidiasis, with the appearance of non-*albicans* isolates, and isolated cases of candidemia in camps following disasters [18,250]. However, these studies tend to focus on urban host populations, leaving out both camp dwellers and migrants in transit [251]. The lack of mycology laboratories in the field and limited access to antifungals hinder timely diagnosis and treatment [176,252].

To fill these gaps, multicenter studies in displaced camps are required that:Quantify the prevalence of candidiasis (mucocutaneous and invasive) and its distribution by species, including resistance profiles.Identify specific risk factors of displacement, such as hygiene conditions, sanitary access barriers, and psychological stress.Evaluate interventions such as improved sanitation, educational programs, and the use of broad-spectrum antifungals in vulnerable populations [125].

To design a comprehensive research plan on candidiasis in displaced populations, six strategic lines are proposed, ranging from prevalence studies and antifungal profiling to clinical trials and the integration of a One Health approach:Multicenter prevalence studiesConduct cross-sectional studies in camps and shelters to quantify the prevalence of oral and vulvovaginal candidiasis, itemized by sociodemographic factors, nutritional status, and HIV co-infection [227].Collect samples with swabs from the oral cavity, vaginal tract, and intertriginous zone, and analyze them by culture and quantitative PCR for the detection of *Candida* [177,253,254].Species identification and antifungal profileEstablish mobile laboratory units or partner with regional diagnostic centers (e.g., hospitals in host cities) to accurately identify *Candida* species and perform MIC determinations using broth microdilution [255].Establish a regional registry of isolates from migrants and displaced persons in LA&C, including genetic typing using MLST (multilocus sequence typing) [256].Study of risk factors associated with displacementDesign cohort studies that evaluate how overcrowding, access to water and hygiene, and dietary patterns influence the development of candidiasis [83,103,114,124,132,195,257].Supplement with qualitative analyses using semi-structured interviews on hygiene and self-care practices implemented within the camps [196,211].Design clinical trials to evaluate preventive and therapeutic interventionsImplement a controlled clinical trial to evaluate the frequency of candidiasis in participants who receive oral and genital hygiene training and basic supplies (inexpensive antifungal soap and hypoallergenic towels) compared to those with conventional management [183,258].Conduct a randomized study in patients with recurrent vulvovaginal candidiasis to compare short treatment regimens with FCZ versus topical therapy with azoles (miconazole), evaluating adherence and tolerance [259].Systematic monitoring in camp healthcare centersImplement active surveillance in mobile hospitals and field clinics to detect fungemia in critically ill patients through local blood cultures and BDG measurement [129,179].Conduct case–control studies in displaced patients with prolonged hospital stays to identify specific risk factors for IC (such as neutropenia, catheter use, or TPN) [80,81,142].Integration of a One Health approachInvestigate the potential transmission of resistant strains of *Candida* through common environments—such as contaminated water sources or food—in camps [1,51,98].Establish partnerships with veterinarians and local bioresource centers to identify environmental reservoirs of *Candida* in temporary settlement sites [106].

## Figures and Tables

**Figure 1 jof-11-00583-f001:**
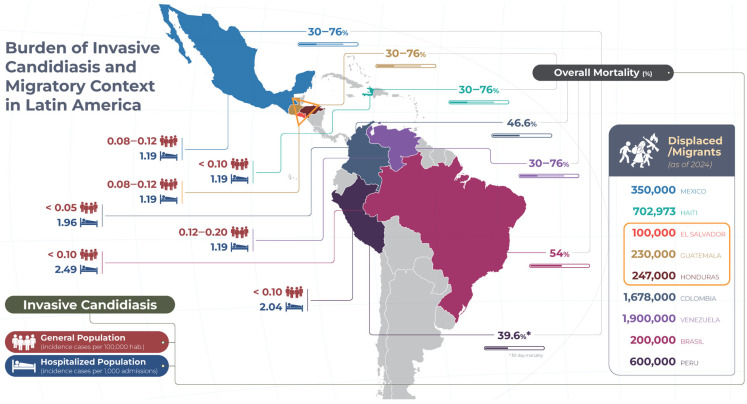
Data boxes for each country show the number of internally displaced persons in 2024 (box size proportional to volume), the incidence of invasive candidiasis in the general population (cases per 100,000 inhabitants) and in hospitalized patients (cases per 1000 admissions), and 30-day overall mortality (%); the right-hand inset summarizes displaced persons by country.

**Table 1 jof-11-00583-t001:** Major armed conflicts in LA&C (2000–2025) [20,21,22,23,24,25].

Country	Type of Conflict/Violence	Key Period	Internally Displaced Persons/Migrants-Refugees (Estimates)	Key Findings
Mexico	Drug trafficking and violence by criminal groups	2006–2020	–/127,796 ^1^	Border areas and informal transit routes: critical points of displacement. A large number of the victims end up seeking asylum or in transit to the U.S.
Haiti	Political instability, gang violence, and natural disasters	2004–2020	702,973/350,000 ^2^	Territorial control by armed gangs causes recurrent displacement. The 2010 earthquake increased vulnerability and hindered sustainable returns.
El Salvador	Gang violence (MS-13, Barrio 18)	2000–2020	513,000/234,000 ^3^	Extortion and targeted killings forced families to flee as a preventive measure. There is a strong correlation between areas of high extortion and irregular migration.
Guatemala	Civil war (1960–1996) and post-conflict violence	1960–1996/2000–2020		Rural massacres displaced indigenous communities. Gangs and organized crime continue to force partial internal exoduses.
Honduras	Gang violence and organized crime	2000–2020		Extortion by gangs: main trigger of forced displacement. Migration to cities with less violence creates new peripheral settlements.
Colombia	Internal armed conflict (FARC, paramilitaries)	2000–2020	6,900,000/3,000,000 + 2,900,000 ^4^	Persistence of rural “confinements” after the peace agreement. Selective violence drives internal migration to medium-sized cities.
Venezuela	Sociopolitical crisis and high levels of street violence	2014–2020	–/7,770,000 ^5^	Economic collapse and failures in basic services: the main cause of internal migration. Urban displacement concentrates vulnerability in the outskirts of Caracas and Maracaibo.
Brazil	Urban violence and organized crime (favelas, militias)	2000–2020	–/790,000 ^6^	Police operations and clashes with militias generate temporary relocations. Intense impact on favelas in Rio de Janeiro and Sao Paulo.
Peru	Internal conflict (Sendero Luminoso (Shining Path) and State response)	1980–2000	–/Not available	Mass displacement of Andean and indigenous communities. The aftermath of State repression maintains social fractures in rural areas.

FARC: Revolutionary Armed Forces of Colombia, UNHCR: United Nations High Commissioner for Refugees. ^1^ No official figures available for internally displaced persons for 2023–2024 (National Victims Commission, UNHCR). ^2^ Record number of internally displaced persons due to criminal violence in mid-2024. ^3^ No official figures for 2024; recent increase documented by regional analysts in Guatemala and Honduras. ^4^ Displacements reported since the signing of the peace agreement (November 2016) until June 2024. ^5^ Latest UNHCR estimate; unpublished internal displacement data for 2023–2024. ^6^ Localized displacement; no recent UNHCR report for 2024.

**Table 2 jof-11-00583-t002:** Health Conditions and IC Risk: General vs. Migrant/Displaced Population [1,50,114,127,132,133,134,135,136,137,138,139,140,141,142,143,144,145,146,147,148,149,150,151,152,153,154,155].

Country	Main Risk Factors for IC	Common Conditions and Fungal Risk General Population	Common Conditions and Fungal Risk Displaced/Migrant Population *	Comments
Mexico	- CVC—Broad-spectrum antibiotics—TPN—Colonization by *Candida*	- Infant malnutrition (13%)—HIV/AIDS (0.3%)—TB (22/100,000) - Fungal risk: 1.19‰ (regional mean). Associated with CVC and TPN	- Malnutrition and anemia in Central American migrants—COPD in rural displaced - Fungal risk: probable aspergillosis and candidemia in detention centers	The high rotation in shelters and the presence of wet soils increase the exposure to fungal spores.
Haiti	- Broad-spectrum antibiotics—CVC—TPN	- Severe malnutrition (22%)—Epidemics of cholera and diarrhea - Fungal risk: post-breakout mucocutaneous mycoses; limited data on IC	- Critical post-disaster malnutrition—Cholera outbreaks in camps - Fungal risk: probable increase in mucocutaneous and systemic candidiasis	Sanitary collapse and lack of available antifungals limit the management of deep infections.
Central America Northern Triangle (El Salvador. Guatemala. Honduras)	- Broad-spectrum antibiotics—CVC—TPN—Invasive procedures	- Chronic malnutrition (25–30%)—TB (≈100/100,000)—Trauma and violence - Fungal risk: skin cysts and candidemia in municipal ICUs	- Malnutrition in rural displaced persons—TB in informal camps - Fungal risk: IC in burns and associated with CVC	The insecurity and dispersion of the population make prevention and timely treatment interventions difficult.
Colombia	- Broad-spectrum antibiotics (OR 5.6)—CVC (OR 4.7)—TPN (OR 4.6)—Colonization by *Candida*	- Chronic malnutrition (10.5%)—HIV/AIDS (0.4%)—TB (21/100,000) - Fungal risk: candidemia associated with CVC. TPN. and antibiotics (OR 5.6)	- Acute and chronic malnutrition—HIV/AIDS (~0.5%)—Elevated risk for pulmonary and extrapulmonary TB - Fungal risk: similar to the general hospitalized. aggravated by overcrowding and malnutrition	The combination of malnutrition and overcrowding enhances immunosuppression. increasing susceptibility to HF.
Venezuela	Same as Colombia. but with scarce local data	- Infant malnutrition (11.3%)—Diabetes (8%)—TB (17/100,000) - Fungal risk: similar to Latin America, with local under-registration	- Malnutrition due to food crisis—Chronic diseases (diabetes, HTN)—TB in returnees - Fungal risk: possible delay in diagnosis, with poor surveillance	The humanitarian crisis hinders timely access to antifungal diagnosis and treatment.
Brazil	- CVC—Broad-spectrum antibiotics—TPN—Recent surgery	- Infant malnutrition (7.4%)—HIV/AIDS (0.4%)—TB (32/100,000) - Fungal risk: outbreaks of candidemia in the ICU; increased isolates of *N. glabratus*	- Moderate malnutrition in Haitian refugees—HIV/AIDS in urban displaced persons—CKD (dialysis) in Haitian refugees—Chronic kidney disease (dialysis) in Haitian refugees - Fungal risk: elevated in dialysis and ICU patients	Migration from Haiti generates social stress and hinders the continuity of antifungal treatments.
Peru	- Broad-spectrum antibiotics—CVC—Recent surgery—MV—TPN	- Infant malnutrition (12%)—TB (119/100 000)—HIV/AIDS (0.3%) - Fungal risk: incidence of 2.04‰ in ICU; mortality 39.6%	- Malnutrition in rural displaced persons—Extrapulmonary TB—TB-HIV co-infection in migrants - Fungal risk: similar to general inpatients	Internal-urban displacement delays seeking care, increasing fungal complications.

* As no specific studies were found in this population, risk factors derived from social determinants of health are proposed. TB: tuberculosis, TPN: total parenteral nutrition, CVC: central venous catheter, MV: mechanical ventilation, IC: invasive candidiasis, COPD: chronic obstructive pulmonary disease, ICU: intensive care unit, HTN: hypertension; CKD: chronic kidney disease.

**Table 3 jof-11-00583-t003:** Gaps in Access to IC Diagnosis and Treatment: General vs. Migrant/Displaced Population [1,12,51,117,175,180,181,182].

Country	IC Burden (Incidence; Mortality)	Diagnosis and Treatment General Population	Diagnosis and Treatment Displaced/Migrant Population	Comments
Mexico	1.19‰; 30–76% (regional mean)	Diagnosis:—Cultures and stains (48–72 h)-BDG/AGA (3 national institutes)-MALDI-TOF (5 high-complexity hospitals) Treatment:—FCZ-Standard AmB—L-AmB (limited)-Echinocandins (available, irregular distribution)	Diagnosis:—Cultures and stains in large hospitals—BDG and MALDI-TOF almost absent at the second level Treatment:—FCZ-D-AmB-Echinocandins (very limited access)	Existing national protocol, but uneven distribution of reagents and drugs delays the management of migrants and displaced persons.
Haiti	1.19‰; 30–76% (regional mean)	Diagnosis:—Cultures in 2 central laboratories (48–72 h)-No BDG, PCR, or MALDI-TOF. Treatment:—Generic AmB—FCZ (ONG)	Diagnosis:—Clinical (lack of specialized laboratories) Treatment:—AmB-FCZ-Echinocandins (not available)	Healthcare collapse and dependence on NGOs leave displaced persons without timely diagnosis and appropriate treatment.
Central America Northern Triangle (El Salvador, Guatemala, Honduras)	1.19‰; 30–76% (regional mean)	Diagnosis:—Local cultures—No BDG, PCR, or MALDI-TOF Treatment:—D-AmB-FCZ-ITZ-Echinocandins (not included in the public service)	Diagnosis:—Clinical and primary cultures—No immunological/molecular testing Treatment:—AmB-FCZ-Echinocandins (not accessible)	Lack of training and equipment in local laboratories; diagnostic delays increase mortality among displaced persons.
Colombia	1.19‰; 30–76% (regional mean)	Diagnosis:—Cultures and stains (48–72 h)-BDG (reference)-MALDI-TOF (large hospitals) Treatment:—D-AmB, L-AmB (limited)-FCZ, ITZ-Echinocandins (expensive, scarce)	Diagnosis: Cultures and stains (reference)—No BDG/MALDI-TOF in primary care Treatment:—AmB-FCZ-Echinocandins (not accessible)	Delays in diagnosis and lack of echinocandins in the public system increase mortality in displaced persons.
Venezuela	1.19‰; 30–76% (regional mean)	Diagnosis:—Cultures (very slow)—No BDG nor MALDI-TOF Treatment:—Generic AmB—FCZ (unstable stock)	Diagnosis:—Predominantly clinical—No immunological/molecular testing Treatment:—AmB-FCZ-Echinocandins (not available)	Chronic shortage of reagents and antifungals aggravates delays in displaced persons.
Brazil	2.49‰; 54%	Diagnosis:—Cultures and stains (48–72 h)—BDG (2–3 reference laboratories)—MALDI-TOF (large hospitals) Treatment:—AmB (including liposomal)—FCZ, VCZ, echinocandins	Diagnosis:—Cultures/stains in urban centers-Little BDG/MALDI-TOF outside of capitals—PCR almost null Treatment:—D-AmB-FCZ-Echinocandins (limited)	Large health centers offer adequate care, but rural refugees and displaced persons are almost completely lacking these options.
Peru	2.04‰; 39.6% (30 days)	Diagnosis:—Cultures (48–72 h)-BDG (2 laboratories in Lima)-MALDI-TOF (1 university center) Treatment:—D-AmB, L-AmB (limited stock)—FCZ, ITZ, echinocandins (irregular stock)	Diagnosis:—Cultures—Outside Lima without BDG/MALDI-TOF/PCR Treatment:—AmB-FCZ-Echinocandins (virtually inaccessible)	Concentration of resources in Lima leaves displaced persons from the interior of the country with almost insurmountable barriers to diagnosis and treatment.

Note: Regional incidences reflect the best available estimate where no specific national data are available. IC: invasive candidiasis; BDG: (1 → 3)-β-D-glycan, PCR—polymerase chain reaction; MALDI-TOF: mass spectrometry, FCZ: fluconazole, ITZ: itraconazole, AmB: Amphotericin, D-AmB: Amphotericin B deoxycholate, L-AmB: Liposomal Amphotericin B, VCZ: voriconazole.

## Data Availability

No new data were created or analyzed in this study. Data sharing is not applicable to this article.

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
