# Peer review of "Invasive Candidiasis in Contexts of Armed Conflict, High Violence, and Forced Displacement in Latin America and the Caribbean (2005–2025)"

_jof, 2025, doi:10.3390/jof11080583_

Round 1
Reviewer 1 Report
This manuscript describes a very comprehensive review of a complex, vast and challenging topic. The manuscript covers an enormous range of information from a host of varied sources. The review attempts to summarize how social upheaval (caused by natural disasters, wars, violence, etc.) has compromised healthcare provision in parts of Latin America and the Caribbean and how this has ultimately affected the incidence of invasive candidiasis (IC) in these areas. Perhaps unsurprisingly, in countries that may already have challenges in providing infrastructure and resources for cutting-edge healthcare, delayed diagnosis and lack of appropriate antifungals can lead to increased incidence of this critically important disease.
The subject area is topical and novel, as there do not appear to have been any previous reviews detailing the epidemiology of IC in these circumstances and in this part of the world. At the end of the review the authors describe a large number of recommendations for future research. However, the problem remains that even in parts of the world that are considered to be stable and developed, optimal outcomes rely on effective surveillance and expensive diagnostic tests to allow timely prescription of expensive drugs. Unfortunately, with the best will in the world, these are unlikely to be available in remote, resource-challenged, mobile hospitals and field clinics.
Although I commend the authors for their comprehensive and thorough review, it is very long indeed. There are parts of the manuscript that are quite repetitive. I believe, the review would achieve more impact if it was significantly edited to remove repetitive details. The review also references mucocutaneous Candida infections throughout. I understand that these need to be referred to, particularly as they no doubt occur in these populations more frequently, however, they sometimes distract from the narrative regarding invasive disease.
In developed countries many patients contract IC because of neutropenia (often due to chemotherapy/transplant therapy) and is mainly a healthcare acquired infection (HAI). Other hospital-related risk factors include TPN and IV catheterization). In the contexts that the authors are describing, where the incidence of IC is even higher than in developed regions, are these also associated with hosptalization, or are they acquired in the community (due to trauma, malnutrition, etc)? Perhaps if there are differences in the underlying causes and risk factors these could be discussed more explicitly in the text?
Invasive candidiasis is often cited as the fourth or fifth most common cause of bloodstream HAIs (usually behind Gram - spp., staphylococci and enterococci). Are there any data on the incidence of these infections in the geographic areas covered by the review?
The authors have been exhaustive in citing a wide range of sources; including primary research papers, reviews, policy documents, newspaper articles, etc. However, these are not always cited correctly/completely in the reference list. For example, ref 1 doesn't have the volume, epage number etc. Is the date cited and "availabe from" really necessary? The authors must comb through the references and ensure all of them are complete and correctly cited according to the journal's requirements as outlined in the Instructions to Authors (i.e. Journal Articles: 1. Author 1, A.B.; Author 2, C.D. Title of the article. Abbreviated Journal Name Year, Volume, page range).
It is also very important that the references cited refer to the relevant information. For instance, are the figures cited in lines 60-61 found in refs 1 and 8?
Line 118 - is there confusion between the content of Table 1 and Figure 1, I think these are the reverse of what is referred to as the former and latter. Although Fig. 1 looks impressive it is quite confusing, the source of data for each point should be included in a complete legend, not in the main text.
Line 272-3 Sentence seems to be duplicated.
Line 310, it might be helpful to explain that N. glabratus was formerly known as Candida glabrata until relatively recently and is still (inaccurately) referred to as such.
Reviewer 2 Report
This article addresses an important scientific and humanitarian topic of invasive candidiasis in areas lacking political ans sociological stability, focusing on LATAM and the Carribeans. The review comprehensives summarizes exisiting scientific knowledge in this area and could serve as an informative and instructive reference for policy making, humanitarian efforts, and global support.
Section 2 starting from page 2: While the history of confliencts in this region is the important background to the issue of invasis candidiasis epidemiology, I would recommend keeping this section short and succinct. The depth of discussion could be better appreciated in a political, histological, or humanitarian discussion instead of JoF. The numbers are certainly shocking, hence I like the use of Figure 1 and Table 2 that summarize and visualize the statistics well.
On Page 8, LIne 283, it seems counter-intuitive that LA&C contributes to less global burden of IC (8%) than the weight of its population (13%), especially considering the instability described earlier in this review. It would be helpful if the authors could justify such disproportion.
On Page 10, line 389, I am not sure what "000 admissions" mean. Is it a typo?
On Page 11, line 469, I assume "factores" is the spanish word not removed after translation. It would be great if the authors could confirm if they did not have other meaning in mind for risk factors.
On Page 18, Table 3, the IC burden is identical for 5 countries/regions, showing 1.19‰; 30-76%. Could the authors correct why these statistics are identical?
